# Elder abuse as a risk factor for psychological distress among older adults in India: a cross-sectional study

Maria Evandrou, Jane C Falkingham, Min Qin, Athina Vlachantoni

## ABSTRACT

**Objectives** This study examines the association between elder abuse and psychological distress among older adults in India and explores whether this association varies by the level of psychosocial and material resources.

**Design** The study uses a cross-sectional survey design.

**Setting** The data are drawn from a representative sample of 9589 adults aged 60 and above in seven Indian states—Himachal Pradesh, Punjab, West Bengal, Odisha, Maharashtra, Kerala and Tamil Nadu—in 2011.

**Statistical analyses** Secondary analysis, using bivariate and multivariate logistic regression models, is conducted using the United Nations Population Fund project Building Knowledge Base on Ageing in India survey. Elder abuse (physical and/or emotional) emanating from family members in the previous month before the survey is examined. Multivariate models are run on the total analytical sample and for men and women separately.

**Results** The overall prevalence of psychological distress among persons aged 60 and over living in the seven Indian States is 40.6%. Among those older persons who experienced some form of physical or emotional abuse or violence in the last month, the prevalence of psychological distress is much higher than that in the general older population, at 61.6% (p<0.001). The results show that the experience of abuse is negatively associated with the mental health of older adults, and this relationship persists even after controlling for demographic and socioeconomic factors (OR=1.60, 95% CI 1.22 to 2.09). The findings also suggest that household wealth has an inverse relationship with mental health, with the association between experiencing elder abuse and reporting poor mental health being strongest among older people in wealthy households.

**Conclusions** Elder abuse in India is currently a neglected phenomenon, and greater recognition of the link between abuse and mental health is critical to improve the well-being of vulnerable older adults, some of whom may be 'hidden' within well-off households.

Centre for Research on Ageing, ESRC Centre for Population Change, University of Southampton, Southampton, UK

**Correspondence to**
Dr Min Qin;
min.qin@soton.ac.uk

## Strengths and limitations of this study

► The findings of this study are from a large representative sample of 9692 older adults from selected Indian states.
► Abuse data in this study are self-reported; there is no validation (or under-reporting) by agencies charged with investigating elder abuse.
► It is possible that older people already had stress before the experience of abuse, and at the time of the interview were more likely than others to recall experiences such as abuse.

## INTRODUCTION

India's population is ageing. With improvements in mortality as a result of rising living standards, improved sanitation, public health and medical advances, more people are living longer and surviving into old age. Such trends, combined with recent falls in fertility, mean that the number of older people is both increasing in absolute terms and also as a share of the population. In 1980, individuals aged 60 and over accounted for just 5.9% of the Indian population; by 2015 this had risen to 8.9%, comprising 116.5 million people, and by 2050 older people are projected to constitute nearly one-fifth (19.4%) of the total population, with 330 million Indians in their 60s or older.[1] The changes in the age structure of India's population are being accompanied by other social and economic transformations including rapid urbanisation and industrialisation. Increasing women's participation in paid employment, greater internal and international mobility among the younger generation and the growth of individualism are all impacting on the traditional Indian family system; a system which has emphasised the obligation of sons and their wives to respect, obey and provide care for their aged parents.[2]

Traditionally, elders have been respected in Indian society, and families are the principal financial, emotional and physical caregivers for older relatives.[3] Although this tradition remains today,[4] qualitative studies have demonstrated that both respect for older people and the caring traditions of the extended family are on the wane in the larger context of societal and cultural changes. Older people are more likely to be exposed to abuse, isolation and abandonment.[5-6] Elder

abuse is internationally defined as a "single, or repeated act, or lack of appropriate action, occurring within any relationship where there is an expectation of trust which causes harm or distress to an older person".[7] Elder abuse is estimated to affect one in six older adults worldwide, has become a growing public health challenge and requires more attention by healthcare systems, researchers and more evidence-based intervention.[8-9] Across a wide range of countries, risk factors for elder abuse include functional dependence or physical disability, poor physical and mental health and low socioeconomic status. Most international studies found women are more likely than men to experience elder abuse.[10]

Several theories may explain the possible causes of elder abuse by family members.[11] According to the social exchange theory, elder abuse may arise because of older people's dependence on the family members, while situational theory focuses on the role of stress and the burden of caregiving as precursors to elder abuse. An overburdened family member who cannot cope with caring demands may create an environment which is conducive to abuse. Symbolic interactionism theory emphasises the role of cultural values and expectations in influencing the perception of elder abuse. For example, in some elders cultural perceptions, going to live in a nursing homes is considered to be a form of abuse, whereas their children may define it as a sign of caring.

Elder abuse can manifest as physical, emotional, sexual and financial abuse and/or as intentional or unintentional neglect. In the Indian context, older people customarily perceive the word 'abuse' to mean extreme behaviour of violence but not neglect/abandonment. However, in qualitative studies, older people have acknowledged the existence of maltreatment (lack of dignified living and disrespect) and neglect within their society and community. In addition, women have been considered as the worst sufferers with no income of their own and being dependent on other family members for everything.[12] Qualitative research has found that selected later life mental disorders may be attributed to abuse, neglect or lack of love from children.[5] However, psychological distress in later life remains an under-researched area in India, particularly in terms of the psychological consequences associated with elder abuse and neglect.[12-13]

### The relationship between elder abuse, resources and psychological distress

Psychological distress is widely used as an indicator of the mental health of the population within the field of public health (psychological distress and poor mental health are interchangeable terms in this paper). Distress comprises a variety of symptoms such as depression, anxiety, stress and insomnia. The level of such distress experienced by an individual at any point in time is determined by various biological and psychosocial factors.[14-15] Elder abuse is recognised as a stressful experience which has been found to have harmful effects on mental health,[16]

with depression, anxiety and post-traumatic disorder being reported as the most prevalent psychological consequences.[17-18]

The general stress theory postulates that the effect of stressors (ie, stressful events) on psychological health operates in three phases: alarm, resistance and exhaustion and is a process that involves changes in individuals' immune system, endocrine system and cardiovascular reactivity.[19] When problems accumulate, persist and strain individuals, then, adaptation resources are depleted and a stimulated parasympathetic system may lead to worrying, anxiety, depression, anger and/or other physical illness. Studies have found that older adults who are mistreated have higher levels of psychological distress than those who have no such experience.[20-21] The frequency or type of elder abuse also has an impact on mental health. Fisher and Regan[22] found that repeated abuse or multiple types of elder abuse (eg, emotional) were risk factors for depression or anxiety among older women. Based on previous empirical findings, this paper hypothesises (H1) that older adults who report experiencing elder abuse will have higher odds of psychological distress than those who do not report such experiences.

However, many individuals who experience stressful events do not go on to develop a psychological illness. The stress-buffering model suggests that psychosocial resources moderate the deleterious effects of high levels of stress. The statistical interaction between stress and resources can be used to test these moderating effects.[23] Resources may intervene between the experience of stress and the onset of mental illness by providing a solution to the problem or reducing the perceived importance of the problem, which in turn helps to decrease or eliminate the stress reaction. Psychosocial resources that can buffer the negative impact of life events on psychological well-being include subjective resources such as high self-esteem, mastery, social support and social participation, and objective resources such as socioeconomic status, including income and household wealth.[24-26] For instance, a beneficial effect of social support on one's mental health could occur, thanks to large social networks providing individuals with regular positive experiences and stable, socially rewarding roles in the community. This kind of support could provide a positive effect, a sense of predictability and stability in one's life situation and recognition of self-worth, all of which are related to overall well-being. At the same time, material resources such as income and household wealth can offer protection against negative experiences associated with economic problems. Indeed, previous empirical studies have found that in the presence of stress from elder abuse, supportive relationships may buffer the effect of stress.[27, 21] As such, this paper hypothesises that (H2) the negative association of elder abuse with psychological distress will be stronger for those with fewer psychosocial and material resources than for those with more psychosocial and material resources. This study aims to contribute to the literature by investigating the association between elder abuse and

psychological distress among older adults in India and examining whether such association varies by the level of psychosocial and material resources at the older adults' disposal.

## DATA AND METHODS

This study analyses data collected as part of the United Nations Population Fund 'Building Knowledge Base on Ageing in India (BKPAI)' project. The BKPAI Survey was conducted in 2011 in seven major demographically advanced states of India—Himachal Pradesh, Punjab, West Bengal, Odisha, Maharashtra, Kerala and Tamil Nadu. A representative sample was obtained using a random sampling method covering the Northern, Southern, Western and Eastern regions. The detailed information about the survey sampling is described in a previous report.[28] The primary sampling units were households. All those aged 60 and above in the sampled households were interviewed face to face. The completion rate for households was 94.7% and 92.9% for elderly respondents. Non-response at both the household and individual levels was adjusted through the sampling weights calculation by the research organisation. The BKPAI survey data include information on older people's mental and physical health, their living arrangements, socioeconomic circumstances, including employment status and household assets, as well as information on intergenerational exchanges within the family and participation in social activities. The total sample size interviewed is 9692. Of these, 103 are excluded because of missing values (missingness is not mutually exclusive) on psychological distress (N=36); education (N=53); whether has someone for trust/confidence (N=7) and whether feels able to manage unexpected situations (N=16). The final analytical sample is 9589 adults aged 60 and above.

## Measurements

### Elder abuse

In the BKPAI survey, the respondents were asked two sets of questions regarding their experience of abuse since they were 60 years old and in the last month. The first question was "In the time since you completed 60 years of age have you faced any type of abuse or violence or neglect or disrespect by any person?" If the respondent answered 'Yes', a follow-up questions asked the type of abuse (physical abuse, verbal abuse, economic abuse, showing disrespect, neglect and other) and where it originated (within family, outside family, both within family and outside family). A further question asked, "Have you faced any type of physical or emotional abuse or violence in the last month?" The responses include: "(1) No; (2) Physical; (3) Emotional; (4) Both physical and emotional". All other types of violence other than physical were merged into emotional violence. If the respondent answered in the affirmative, follow-up questions elicited the source of abuse which could include: "(1) Spouse; (2) Son; (3) Daughter; (4) Son-in-law; (5) Daughter-in-law;

(6) Domestic helper; (7) Grandchildren; (8) Relatives; (9) Neighbours; (10) Other".

A previous study based on this data reported that 11% of respondents have experienced at least one type of abuse after the age of 60. Verbal abuse is most frequently claimed, followed by disrespect, economic abuse, neglect and physical abuse, with the most common perpetrator being the respondent's son.[17] In this study, we concentrate on older adults who report having experienced physical and/or emotional abuse in the last month, distinguishing between those who report abuse by family members and others, to examine the contemporaneous interaction between elder abuse, psychosocial and material resources and psychological distress. Here, abuse is limited to that reported as emanating from family members as it is this form of abuse that we hypothesise may have increased as a result of recent changes impacting the traditional Indian family system.

### Psychological distress

The 12-item version of the General Health Questionnaire (GHQ-12) is used as a measure of psychological distress. These questions have been widely used to identify minor psychiatric disorders in the general population.[29] The GHQ-12 has been previously validated in India in clinical surveys conducted in Kannada,[30-31] Hindi[32] and Tamil.[33] Given that the BKPAI study was conducted across multiple states with different languages and focused among older adults, it was important for the team to test the reliability of GHQ-12 within the BKPAI. The measure was found to have high internal consistency with an overall Cronbach's alpha of 0.9. Examining each state individually, Cronbach's alpha ranged from a low of 0.7 in West Bengal to a high of 0.94 in Himachal Pradesh, suggesting that the measure may be considered to be valid across all seven states and in all the languages used.

Using the standard GHQ scoring method, the four category responses for each of the 12 questions are coded (0, 0, 1, 1), with the points summed to produce a total score ranging from 0 to 12. We used a score of >=4 as the threshold to define psychological distress according to studies validating the GHQ-12 against standardised psychiatric interviews.[34, 31] Although a Likert scale (0-1-2-3) scoring method is also widely used, a previous study found that for the GHQ-12, the GHQ scoring method was more effective than the Likert method when defining the distressed cases.[29]

### Psychosocial and material resources

An individual's psychosocial resources include personal qualities such as optimism, psychological control or mastery and self-esteem, as well as the availability of social support, all of which can help to manage stressful events and contribute to better health outcomes.[35,24,26] There are a variety of scales measuring social support and personal coping resources.[36-37] The BKPAI was not explicitly designed to measure psychosocial resources. It does, however, contain a number of important indicators

of potential support and coping resources, including being married or living together with one's partner, participation in social activities, having someone to trust and confide in and feeling able to manage unexpected situations.

Participation in social activities is defined as having participated in any of the five listed activities in the last 12 months: attending a public meeting with discussion of local, community or political affairs; attending any group, club, society, union or organisational meeting; working with other people in your neighbourhood to fix or improve something; attending or participating in any religious programmes/services (not including weddings and funerals); going out of the house to visit friends or relatives.

The question on feeling able to manage unexpected situations has three response categories: most of the time, sometimes and hardly ever feeling that one can manage situations even when they do not turn out to be as expected.

Material resources include personal financial dependency (no dependency, partial dependency, full dependency) and household wealth quintile index. Household wealth quintile index is computed using principle component analysis based on 30 assets and housing characteristics: household electrification; drinking water source; type of toilet facility; type of house; cooking fuel; house ownership; ownership of a bank or postoffice account and ownership of a mattress, pressure cooker, chair, cot/bed, table, electric fan, radio/transistor, black and white television, colour television, sewing machine, mobile telephone, any landline phone, computer, internet facility; refrigerator, watch or clock, bicycle, motorcycle or scooter, animal-drawn cart, car, water pump, thresher and tractor. This measure was found to provide a good socioeconomic gradient of health outcomes among older adults in the survey. [28]

### Other control variables

Covariates include the individual's age group, sex, education, caste, working status, living arrangement, self-reported health, chronic disease, health-related limitations to daily activities, disability and geographic factors (rural/urban residence and state). Functionality is measured using two derived variables capturing (1) an individual's reported need for assistance with Activities of Daily Living (ADL) and (2) Disability. ADLs refer to the ability to perform basic daily activities; disability is associated with a decline of motor function. In general, a high score of disability represents a lower ability to perform ADLs. [38] ADL is computed based on the level of independence reported by the older person in carrying out the activities of feeding, bathing, dressing, toilet, mobility and continence. Each question has three response categories: 'Do not require assistance; Require partial assistance; Require full assistance'. These are scored as 0, 1 or 2, respectively, and are then summed across the six questions, resulting in a total score ranging between 0 and 12. Given the

unequal intervals between the score, rather treating it as a continuous variable, we group it into an ordered categorical variable. Older respondents are defined as having 'no need for assistance' if the total score is 0, as having a 'light need for assistance' if the total score is between 1 and 5 and as having a 'heavy need for assistance' if the total score is ≥ 6.

Disability is computed based on the respondents' level of reported ability to see, hear, walk, chew, speak and remember. Each question has three response categories 'Yes fully, Yes partially, No'. These are scored as 0, 1 or 2, respectively, and are then summed across the six questions, resulting in a total score of 0-12. Again, because of the unequal intervals between the score, we group the total score into an ordered categorical variable. Older people are defined as having 'no disability' if the total score is 0, 'light disability' if the total score is between 1 and 2, 'medium disability' with a score of 3-4 and 'high disability' if the total score is ≥ 5.

### Statistical analyses

The bivariate associations of psychological distress with exposures and potential risk factors are explored using the $\chi^2$ test first. Then, a series of logistic regression models are estimated with the dependent variable being the report of psychological distress (GHQ ≥4 contrast to GHQ ≤3). The first model estimates the bivariate association between elder abuse and psychological distress. The second model adds the measures of psychosocial and material resources and other control variables to estimate the association of elder abuse and psychosocial and material resources, with psychological distress after controlling for other covariates. The final model includes the interaction terms of elder abuse with psychosocial and material resources variables to the main-effects-only model to examine whether psychosocial and material resources buffer the association between elder abuse and psychological distress. The logistic regression models are run for the total sample and then separately for older men and women using the Statistic software STATA V.12. The significance level is set to 5% (P<0.05).

### Ethics approval

Ethical approval for this study, involving secondary data analyses, has been obtained from the Ethics Committee in the University of Southampton. The survey report and a previous study show that informed consent was obtained from all individuals prior to participation in the primary data collection exercise. Careful attention was paid to avoid the presence of any family members during the collection of data concerning elder abuse and to guarantee the anonymity of all participants and the confidentiality of information. [13, 28]

## RESULTS

### Descriptive findings

Table 1 presents descriptive statistics for the total analytical sample. The overall prevalence of psychological distress among persons aged 60 and over living in the seven Indian States is 40.6%. Around 5% of older adults had experienced some form of physical or emotional abuse or violence in the last month. Among this subgroup, the prevalence of psychological distress is much higher than in the general older population, at 61.6% (p<0.001).

The indicators of psychosocial resources and socio-economic status appear to have an inverse relationship with psychological distress, with those older people living in households in the highest (richest) wealth quintile having a prevalence rate of 21.4% compared with 64.1% among those living in households in the lowest (poorest) wealth quintile. Similarly, those who participated in social activities in the last month are less likely to experience psychological distress than those who did not (37% vs 55.8%, respectively).

Indicators of health status including fair/poor self-related health, heavy difficulty with ADLs and disability all show a positive association with psychological distress. Older people living with their spouse only experience the lowest prevalence of psychological distress (34.9%), while those living alone show the highest prevalence (50.9%). Levels of psychological distress increase with age and are higher among older women (44.7%) than older men (36.2%). One's place of residence seems to play an important role with elders living in urban areas having a lower level of psychological distress than their rural counterparts (35.1% vs 45.7%). There are also considerable interstate variations in the prevalence of psychological distress. A relatively low level of psychological distress is found among older adults in Punjab (20.8%) and Himachal Pradesh (23.5%), contrasting with much higher levels in West Bengal (60.5%) and Odisha (55.5%).

### Multivariate analysis results

Table 2 shows the ORs from the logistic regression models. Among the total sample, model 1 shows the simple bivariate relationships between elder abuse and psychological distress. The OR of 2.44 suggests that older adults who experienced abuse during last month are more than twice as likely to report psychological distress than those with no such experience. This effect is attenuated once psychosocial resources and other control variables are added (OR=1.60, 95% CI 1.22 to 2.09) (model 2). Social activity participation, social support (having someone to trust) and mastery (feeling able to manage situations) are all associated with psychological distress in the expected negative direction. The exception is marital status, with the results indicating that older people who are currently married or living together with partners are more likely to have psychological distress than those who are widowed, although this finding is not statistically significant. Household wealth has an inverse relationship

with psychological distress. Older people's psychological distress is also related to the levels of physical health and different geographic areas. For instance, older people with poor self-rated health are more likely to have psychological distress than those with good health (OR=3.83, 95% CI 3.24 to 4.52); older people living in Tamil Nadu are more likely to have psychological distress than those living in Himachal Pradesh (OR=3.81, 95% CI 3.01 to 4.81). Finally, model 3 presents interactions between the experience of elder abuse and psychosocial factors (trust and mastery) and material resources (education and wealth quintile). The inclusion of interactions adds significant explanatory value to the model with a likelihood ratio test p value of 0.008. The results indicate that psychosocial resources only have a direct negative association with psychological distress; the interaction terms between elder abuse and psychosocial resources variables are not statistically significant. Interestingly, however, positive and significant interactions are observed between the experience of elder abuse and the respondents' household wealth quintile (OR=2.96, 4.37 and 4.57 for the abuse among middle, fourth and highest quintile).

The separate models by gender show similar patterns of bivariate relationships between elder abuse and psychological distress (for men OR=2.14, 95% CI 1.58 to 2.90; for women, OR=2.60, 95% CI 2.00 to 3.39). Interestingly among older men, the significant association disappear once psychosocial resources and other control variables are added, while among older women the association is attenuated but still significant (OR=1.76, 95% CI 1.23 to 2.50). The results of interactions between the experience of elder abuse and psychosocial factors and material resources among women show similar patterns with the total sample. Figure 1 shows the predicted probabilities for GHQ-12 ≥ 4 at each household wealth quintile according to elder abuse experience based on the coefficients from model 3 among the total sample. The chart shows that for those who did not experience elder abuse in the last month, the probability of psychological distress decreases with the increase of household wealth. However, among those who had experienced abuse, the opposite is found, with the probability of psychological distress increasing as household wealth rises.

## DISCUSSION

The analyses in this paper suggest that elder abuse has a significant negative association with the mental health of older Indians. The results support the first hypothesis outlined in this paper. Elder abuse may be thought of as a particularly stressful event in later life. Typically, Indian parents have continued to invest in their children into adulthood and traditionally have expected to be cared for at an older age. If their investment is not reciprocated, their life is likely to be coloured by a sense of injustice and exploitation,[16] which may lead to certain negative effects such as anger, depressed mood and loneliness.[39] Constant negative effects are known to be compromising

**Table 1** Distribution of GHQ-12≥4 (unweighted data)

| Variables | Distribution (%) | Number of respondents | % of GHQ-12 score ≥ 4 | p Value (Pearson $\chi^2$ test) |
|---|---|---|---|---|
| Total | 100.0 | 9589 | 40.6 | |
| Experience abuse last month | | | | |
| No | 95.5 | 9157 | 39.7 | 0.000 |
| Yes | 4.5 | 432 | 61.6 | |
| Selected psychosocial and material resources variables | | | | |
| Marital status | | | | |
| Widowed | 40.5 | 5710 | 48.0 | 0.000 |
| Currently married/ living together | 59.5 | 3879 | 35.7 | |
| Social activities | | | | |
| No listed social activity | 19.4 | 1860 | 55.8 | 0.000 |
| Have social activity | 80.6 | 7729 | 37.0 | |
| Have someone trust or confide | | | | |
| No | 17.1 | 1642 | 59.7 | 0.000 |
| Yes | 82.9 | 7947 | 36.7 | |
| Feel able to manage situations | | | | |
| Hardly ever | 24.0 | 2298 | 76.3 | 0.000 |
| Sometimes | 63.2 | 6057 | 32.8 | |
| Most of the time | 12.9 | 1234 | 12.9 | |
| Financial dependency | | | | 0.000 |
| No dependency | 25.3 | 2427 | 25.2 | |
| Partial dependency | 24.6 | 2357 | 40.9 | |
| Full dependency | 50.1 | 4805 | 48.3 | |
| Household wealth index | | | | |
| Lowest | 20.0 | 1915 | 64.1 | 0.000 |
| Second | 20.4 | 1958 | 48.4 | |
| Middle | 19.6 | 1884 | 40.4 | |
| Fourth | 19.8 | 1901 | 28.8 | |
| Highest | 20.1 | 1931 | 21.4 | |
| Other control variables | | | | |
| Age | | | | |
| 60-69 | 63.4 | 6082 | 35.6 | 0.000 |
| 70-79 | 26.4 | 2533 | 46.9 | |
| 80+ | 10.2 | 974 | 55.9 | |
| Gender | | | | |
| Men | 47.4 | 4543 | 36.2 | 0.000 |
| Women | 52.6 | 5046 | 44.7 | |
| Education | | | | |
| None | 46.1 | 4422 | 52.1 | 0.000 |
| 1-4 years | 12.9 | 1241 | 45.2 | |
| 5-7 years | 13.5 | 1297 | 36.9 | |

Continued

**Table 1** Continued

| Variables | Distribution (%) | Number of respondents | % of GHQ-12 score ≥ 4 | p Value (Pearson $\chi^2$ test) |
|---|---|---|---|---|
| 8+ years | 27.4 | 2629 | 21.0 | |
| Caste | | | | |
| Scheduled tribe/ Scheduled caste | 24.2 | 2316 | 47.4 | 0.000 |
| Other backward caste | 34.1 | 3274 | 42.9 | |
| Others | 39.1 | 3753 | 33.4 | |
| Unknown | 2.6 | 246 | 58.5 | |
| Working status | | | | |
| Has never worked | 36.2 | 3472 | 41.2 | 0.000 |
| Has ever worked but not now | 40.7 | 3904 | 42.6 | |
| Has ever worked and is now working | 23.1 | 2213 | 36.2 | |
| Self-reported health | | | | |
| Excellent /Very good | 16.2 | 1557 | 19.1 | 0.000 |
| Good | 30.0 | 2875 | 28.1 | |
| Fair | 36.5 | 3499 | 48.7 | |
| Poor | 17.3 | 1658 | 65.6 | |
| Chronic disease | | | | |
| No | 35.3 | 3389 | 35.3 | 0.000 |
| 1 type | 32.2 | 3087 | 41.7 | |
| 2 more types | 32.5 | 3113 | 45.4 | |
| Difficulty with ADLs | | | | |
| No need for assistance | 92.7 | 8890 | 38.1 | 0.000 |
| Light need | 3.8 | 369 | 67.8 | |
| Heavy need | 3.4 | 330 | 79.1 | |
| Disability | | | | |
| No disability | 27.2 | 2606 | 25.1 | 0.000 |
| Light | 44.7 | 4285 | 37.6 | |
| Medium | 18.6 | 1788 | 54.5 | |
| Heavy | 9.5 | 910 | 72.0 | |
| Living arrangement | | | | |
| Alone | 6.3 | 605 | 50.9 | 0.000 |
| Spouse only | 14.9 | 1432 | 34.9 | |
| At least one child | 71.3 | 6833 | 40.5 | |
| Others | 7.5 | 719 | 44.6 | |
| Residence | | | | |
| Rural | 52.2 | 5001 | 45.7 | 0.000 |
| Urban | 47.8 | 4588 | 35.1 | |
| State | | | | |
| Himachal Pradesh | 15.0 | 1440 | 23.5 | 0.000 |
| Punjab | 13.1 | 1255 | 20.8 | |
| West Bengal | 13.2 | 1263 | 60.5 | |

Table 1    Continued

| Variables | Distribution (%) | Number of respondents | % of GHQ-12 score ≥ 4 | p Value (Pearson $\chi^2$ test) |
|---|---|---|---|---|
| Odisha | 15.3 | 1467 | 55.5 | |
| Maharashtra | 14.6 | 1399 | 44.3 | |
| Kerala | 14.0 | 1341 | 28.0 | |
| Tamil Nadu | 14.9 | 1424 | 50.8 | |

Source: Authors' analysis of UNFPA Building Knowledge Base on Ageing in India 2011 survey.
ADL, Activities of Daily Living; GHQ-12, 12-item version of the General Health Questionnaire; UNFPA, United Nations Population Fund.

to both physical and mental health, with the mechanism of pathogenesis operating through physiological changes, including one's immune suppression and cardiovascular and endocrine reactivity.[19,40] The results suggest that women are more vulnerable than men when encountering abusive behaviours from family members. This might be because those women have fewer psychosocial resources[12] to cope with the negative environment or needed medical assistance as a result of the abuse. Our results are consistent with other empirical studies, suggesting that there is a harmful link between older abuse and psychological health.[27,41,20,21] However, no evidence from this study is found for buffering effects of psychosocial resources, such as social support and perceived ability to control outcomes. The results from this study only highlight a direct and beneficial association between psychosocial resources and psychological health, irrespective of the presence of elder abuse. One possible explanation is that elder abuse is in direct conflict with Indian cultural values, and thus older adults who have been abused may not disclose this information or seek support due to a sense of shame and/or a fear of stigmatisation.[42-43] Another explanation may lie in the scale of the outcome variable. In this study, we focus on psychological distress using a nominal scale measurement. Previous studies that have demonstrated the buffering effects of psychosocial resources have measured psychological distress as a ratio scale,[27,21] and thus it is possible that by using a nominal scale, we may be missing some of the nuances around buffering effects. Interestingly, the results in this paper show that household wealth has a direct and inverse relationship with psychological distress and offers a substantial link with the relationship between elder abuse and psychological distress. Both qualitative and quantitative studies have found that individuals who have financial or physical assets may feel more in control of their lives, leading to less vulnerability to anxiety or mood disorders or less severe psychological symptoms.[12,44] Unexpectedly, however, we found that the negative association between elder abuse and mental health is significantly stronger among older people living in wealthier households. One possible explanation might be that issues of control over property, finance and other decisions may result in more family conflict between parents and their adult children or other relatives among wealthier households than in poorer households. This is consistent with qualitative studies in India which have

highlighted bitter battles in village families between elders and adult children over land and money.[12,6] Adult children have been reported at times to resent the expense of medical care and treatments for their aged parents, especially when some of the children felt they were providing more than their fair share of the total cost. Again, this may be more commonplace among wealthier households, where private medical care is an option. Our findings reinforce previous research demonstrating the role of socioeconomic circumstances in determining older people's mental health. Poor social and economic circumstances affect individuals' health throughout life.[45-47] The results from this study also add to the evidence base with regard to inequalities in older people's mental health related to levels of physical health[48-51] and different geographic areas, reflecting differences in their social, political, economic arrangements and levels of public health services and social protection.[52]

## LIMITATIONS
The present study is limited by the cross-sectional design of the data. It is possible that older people already had stress before the experience of abuse, and at the time of the interview were more likely than others to recall past experiences such as abuse. The results could benefit from repeated measures of psychological distress before and after abusive exposures. Another limitation is the self-report nature of the data. There is no validation by agencies charged with investigating elder abuse. Due to social taboo, elder abuse might be under-reported; meantime, the interviewer might make an educated guess concerning the presence or absence of physical, emotional and financial abuse or neglect by family members, which may bias our results on the relationship between abuse and distress. Future surveys need to develop appropriate screening and assessment tools to identify elder abuse.[11] Our data also lack purposively designed scales measuring social support and personal coping resources. Future research addressing these issues will improve our understanding of the relationship between elder abuse, psychosocial resources and psychological distress.

## CONCLUSION
In conclusion, the findings presented in this paper demonstrate that (1) in the seven Indian states

**Table 2** ORs for GHQ-12 score ≥ 4 (N=9589)

| Variables | Total | | | Men | | | Women | | |
|---|---|---|---|---|---|---|---|---|---|
| | Model 1 | Model 2 | Model 3 | Model 1 | Model 2 | Model 3 | Model 1 | Model 2 | Model 3 |
| Experienced abuse in last month | | | | | | | | | |
| No (ref) | | | | | | | | | |
| Yes | 2.44*** (2.00 to 2.97) | 1.60*** (1.22 to 2.09) | 1.12 (0.53 to 2.36) | 2.14*** (1.58 to 2.90) | 1.42 (0.93 to 2.16) | 0.84 (0.22 to 3.16) | 2.60*** (2.00 to 3.39) | 1.76** (1.23 to 2.50) | 1.33 (0.52 to 3.38) |
| Psychosocial and material resources | | | | | | | | | |
| Marital status | | | | | | | | | |
| Widowed (ref) | | | | | | | | | |
| Currently married/living together | | 1.12 (0.98 to 1.28) | 1.12 (0.98 to 1.28) | | 1.29* (1.02 to 1.63) | 1.30 (1.03 to 1.64) | | 1.06 (0.89 to 1.27) | 1.07 (0.90 to 1.27) |
| Social activities participation | | | | | | | | | |
| No (ref) | | | | | | | | | |
| Yes | | 0.85* (0.74 to 0.98) | 0.85* (0.74 to 0.97) | | 1.03 (0.83 to 1.30) | 1.04 (0.83 to 1.30) | | 0.77** (0.65 to 0.91) | 0.76** (0.63 to 0.90) |
| Have someone trust or confide | | | | | | | | | |
| No (ref) | | | | | | | | | |
| Yes | | 0.69*** (0.60 to 0.80) | 0.70*** (0.60 to 0.81) | | 0.65*** (0.52 to 0.82) | 0.67*** (0.53 to 0.84) | | 0.71*** (0.59 to 0.86) | 0.71*** (0.58 to 0.86) |
| Feel able to manage situations | | | | | | | | | |
| Hardly ever (ref) | | | | | | | | | |
| Sometimes | | 0.25*** (0.22 to 0.28) | 0.25*** (0.22 to 0.28) | | 0.25*** (0.21 to 0.31) | 0.25*** (0.21 to 0.31) | | 0.24*** (0.20 to 0.28) | 0.24*** (0.20 to 0.28) |
| Most of the time | | 0.12*** (0.09 to 0.15) | 0.12*** (0.09 to 0.15) | | 0.12*** (0.09 to 0.16) | 0.11*** (0.08 to 0.16) | | 0.11*** (0.08 to 0.15) | 0.12*** (0.09 to 0.16) |
| Financial dependency | | | | | | | | | |
| No dependency (ref) | | | | | | | | | |
| Partial dependency | | 1.07 (0.91 to 1.25) | 1.08 (0.92 to 1.26) | | 1.05 (0.86 to 1.29) | 1.06 (0.86 to 1.30) | | 1.03 (0.80 to 1.33) | 1.04 (0.81 to 1.35) |
| Full dependency | | 1.31*** (1.12 to 1.53) | 1.31** (1.12 to 1.53) | | 1.57*** (1.26 to 1.95) | 1.58*** (1.26 to 1.97) | | 1.11 (0.88 to 1.40) | 1.12 (0.89 to 1.42) |
| Household wealth index | | | | | | | | | |
| Lowest (ref) | | | | | | | | | |
| Second | | 0.83* (0.70 to 0.97) | 0.80* (0.68 to 0.95) | | 0.77* (0.61 to 0.99) | 0.74* (0.58 to 0.96) | | 0.88 (0.70 to 1.10) | 0.86 (0.68 to 1.08) |

Continued

**Table 2** Continued

| Variables | Total | | | Men | | | Women | | |
|---|---|---|---|---|---|---|---|---|---|
| | Model 1 | Model 2 | Model 3 | Model 1 | Model 2 | Model 3 | Model 1 | Model 2 | Model 3 |
| Middle | | 0.78** (0.65 to 0.94) | 0.73*** (0.61 to 0.88) | | 0.97 (0.74 to 1.27) | 0.90 (0.68 to 1.19) | | 0.68** (0.53 to 0.87) | 0.64*** (0.49 to 0.82) |
| Fourth | | 0.62*** (0.51 to 0.76) | 0.57*** (0.47 to 0.70) | | 0.65** (0.48 to 0.87) | 0.60*** (0.45 to 0.82) | | 0.63*** (0.48 to 0.83) | 0.57*** (0.43 to 0.76) |
| Highest | | 0.54*** (0.43 to 0.67) | 0.50*** (0.39 to 0.62) | | 0.54*** (0.38 to 0.76) | 0.51*** (0.36 to 0.72) | | 0.57*** (0.42 to 0.78) | 0.52*** (0.38 to 0.71) |
| Other control variables | | | | | | | | | |
| Age | | | | | | | | | |
| 60-69 (ref) | | | | | | | | | |
| 70-79 | | 1.20** (1.06 to 1.36) | 1.20** (1.06 to 1.36) | | 1.14 (0.95 to 1.38) | 1.14 (0.95 to 1.38) | | 1.25* (1.06 to 1.48) | 1.26** (1.06 to 1.49) |
| 80+ | | 1.24* (1.03 to 1.50) | 1.25* (1.04 to 1.51) | | 1.11 (0.83 to 1.48) | 1.13 (0.84 to 1.51) | | 1.31* (1.02 to 1.69) | 1.32* (1.03 to 1.70) |
| Gender | | | | | | | | | |
| Men (ref) | | | | | | | | | |
| Women | | 1.12 (0.95 to 1.32) | 1.14 (0.97 to 1.34) | | | | | | |
| Education | | | | | | | | | |
| No schooling (ref) | | | | | | | | | |
| 1–4 years | | 0.88 (0.75 to 1.04) | 0.88 (0.76 to 1.04) | | 0.84 (0.66 to 1.08) | 0.84 (0.66 to 1.08) | | 0.88 (0.70 to 1.09) | 0.88 (0.71 to 1.10) |
| 5–7 years | | 0.86 (0.73 to 1.01) | 0.86 (0.73–1.01) | | 0.87 (0.69–1.11) | 0.87 (0.69–1.11) | | 0.84 (0.67–1.06) | 0.85 (0.68–1.08) |
| 8+ years | | 0.59*** (0.50 to 0.69) | 0.60*** (0.51 to 0.71) | | 0.52*** (0.42 to 0.66) | 0.53*** (0.42 to 0.67) | | 0.64*** (0.50 to 0.83) | 0.66*** (0.51 to 0.85) |
| Caste | | | | | | | | | |
| Scheduled tribe/Scheduled caste (ref) | | | | | | | | | |
| Other backward Caste | | 0.91 (0.79 to 1.06) | 0.91 (0.79 to 1.06) | | 0.99 (0.80 to 1.23) | 0.99 (0.79 to 1.23) | | 0.86 (0.70 to 1.05) | 0.86 (0.70 to 1.05) |
| Others | | 1.05 (0.91 to 1.21) | 1.06 (0.92 to 1.22) | | 1.06 (0.85 to 1.31) | 1.06 (0.86 to 1.32) | | 1.05 (0.87 to 1.28) | 1.06 (0.87 to 1.29) |
| Unknown | | 1.09 (0.78 to 1.53) | 1.08 (0.77 to 1.52) | | 1.23 (0.72 to 2.09) | 1.22 (0.72 to 2.08) | | 1.03 (0.65 to 1.61) | 1.03 (0.65 to 1.62) |
| Working status | | | | | | | | | |

Continued

**Table 2** Continued

| Variables | Total | | | Men | | | Women | | |
|---|---|---|---|---|---|---|---|---|---|
| | Model 1 | Model 2 | Model 3 | Model 1 | Model 2 | Model 3 | Model 1 | Model 2 | Model 3 |
| Not work (ref) | | | | | | | | | |
| Ever work but not now | | 1.21* (1.02 to 1.42) | 1.23* (1.04 to 1.45) | | 1.20 (0.59 to 2.45) | 1.20 (0.59 to 2.46) | | 1.20 (0.99 to 1.46) | 1.22* (1.01 to 1.49) |
| Ever work and now | | 1.17 (0.96 to 1.42) | 1.19 (0.97 to 1.45) | | 1.16 (0.56 to 2.41) | 1.15 (0.55 to 2.40) | | 1.11 (0.85 to 1.45) | 1.16 (0.89 to 1.51) |
| Self-reported health | | | | | | | | | |
| Excellent/very good/good (ref) | | | | | | | | | |
| Fair | | 2.09*** (1.85 to 2.35) | 2.09*** (1.86 to 2.36) | | 2.10*** (1.76 to 2.51) | 2.10*** (1.76 to 2.51) | | 2.08*** (1.77 to 2.45) | 2.09*** (1.78 to 2.46) |
| Poor | | 3.83*** (3.24 to 4.52) | 3.84*** (3.25 to 4.54) | | 4.56*** (3.50 to 5.94) | 4.55*** (3.49 to 5.93) | | 3.47*** (2.78 to 4.33) | 3.50*** (2.80 to 4.36) |
| Chronic disease | | | | | | | | | |
| No (ref) | | | | | | | | | |
| One type | | 1.17* (1.03 to 1.34) | 1.17* (1.03 to 1.34) | | 1.29* (1.06 to 1.58) | 1.29* (1.06 to 1.57) | | 1.11 (0.93 to 1.32) | 1.11 (0.93 to 1.32) |
| More than one types | | 1.31*** (1.13 to 1.51) | 1.31*** (1.13 to 1.51) | | 1.43*** (1.14 to 1.78) | 1.44*** (1.15 to 1.80) | | 1.23* (1.01 to 1.49) | 1.23* (1.01 to 1.49) |
| Difficulty with ADLs | | | | | | | | | |
| No need for assistance (ref) | | | | | | | | | |
| Light need | | 1.21 (0.92 to 1.60) | 1.22 (0.93 to 1.61) | | 0.85 (0.55 to 1.31) | 0.85 (0.55 to 1.31) | | 1.52* (1.06 to 2.19) | 1.53* (1.06 to 2.20) |
| Heavy need | | 1.83*** (1.31 to 2.56) | 1.84*** (1.32 to 2.58) | | 2.75*** (1.51 to 5.00) | 2.79*** (1.54 to 5.08) | | 1.44 (0.95 to 2.19) | 1.46 (0.96 to 2.21) |
| Disability | | | | | | | | | |
| No disability (ref) | | | | | | | | | |
| Light | | 1.55*** (1.35 to 1.78) | 1.54*** (1.34 to 1.77) | | 1.27* (1.03 to 1.56) | 1.25* (1.02 to 1.54) | | 1.79*** (1.49 to 2.16) | 1.79*** (1.48 to 2.16) |
| Medium | | 2.23*** (1.87 to 2.65) | 2.22*** (1.87 to 2.64) | | 1.81*** (1.39 to 2.36) | 1.79*** (1.37 to 2.34) | | 2.56*** (2.03 to 3.23) | 2.57*** (2.03 to 3.24) |
| Heavy | | 3.39*** (2.71 to 4.25) | 3.38*** (2.70 to 4.23) | | 2.69*** (1.92 to 3.77) | 2.67*** (1.90 to 3.75) | | 4.01*** (2.95 to 5.44) | 4.03*** (2.96 to 5.48) |
| Living arrangement | | | | | | | | | |
| Alone (ref) | | | | | | | | | |

Continued

**Table 2** Continued

| Variables | Total | | | Men | | | Women | | |
|---|---|---|---|---|---|---|---|---|---|
| | Model 1 | Model 2 | Model 3 | Model 1 | Model 2 | Model 3 | Model 1 | Model 2 | Model 3 |
| Spouse only | | 0.78 (0.60 to 1.03) | 0.78 (0.59 to 1.03) | | 0.65 (0.37 to 1.13) | 0.66 (0.38 to 1.16) | | 0.82 (0.57 to 1.17) | 0.81 (0.57 to 1.17) |
| At least one child | | 1.10 (0.87 to 1.40) | 1.10 (0.86 to 1.39) | | 0.91 (0.53 to 1.56) | 0.93 (0.54 to 1.59) | | 1.14 (0.86 to 1.50) | 1.14 (0.86 to 1.50) |
| Others | | 1.20 (0.89 to 1.61) | 1.19 (0.89 to 1.60) | | 1.20 (0.66 to 2.20) | 1.22 (0.67 to 2.24) | | 1.13 (0.79 to 1.60) | 1.13 (0.79 to 1.60) |
| Residence | | | | | | | | | |
| Rural (ref) | | | | | | | | | |
| Urban | | 0.90 (0.80 to 1.01) | 0.90 (0.80 to 1.01) | | 0.77** (0.64 to 0.91) | 0.76** (0.64 to 0.91) | | 0.99 (0.85 to 1.15) | 0.99 (0.84 to 1.15) |
| State | | | | | | | | | |
| Himachal Pradesh (ref) | | | | | | | | | |
| Punjab | | 0.60*** (0.48 to 0.74) | 0.60*** (0.48 to 0.75) | | 0.61** (0.43 to 0.86) | 0.62** (0.44 to 0.87) | | 0.56*** (0.42 to 0.75) | 0.56*** (0.42 to 0.75) |
| West Bengal | | 2.63*** (2.13 to 3.25) | 2.60*** (2.10 to 3.21) | | 4.23*** (3.06 to 5.85) | 4.25*** (3.07 to 5.88) | | 1.78*** (1.34 to 2.38) | 1.74*** (1.30 to 2.33) |
| Odisha | | 2.82*** (2.28 to 3.49) | 2.78*** (2.24 to 3.44) | | 3.27*** (2.37 to 4.50) | 3.28*** (2.37 to 4.53) | | 2.65*** (1.98 to 3.56) | 2.58*** (1.92 to 3.47) |
| Maharashtra | | 1.69*** (1.36 to 2.10) | 1.67*** (1.35 to 2.08) | | 2.16*** (1.56 to 2.98) | 2.17*** (1.57 to 2.99) | | 1.37* (1.01 to 1.85) | 1.34 (0.99 to 1.81) |
| Kerala | | 0.96 (0.76 to 1.20) | 0.95 (0.76 to 1.20) | | 0.72 (0.50 to 1.04) | 0.73 (0.50 to 1.05) | | 1.05 (0.78 to 1.41) | 1.03 (0.77 to 1.40) |
| Tamil Nadu | | 3.81*** (3.01 to 4.81) | 3.75*** (2.97 to 4.74) | | 4.63*** (3.24 to 6.59) | 4.62*** (3.24 to 6.60) | | 3.17*** (2.31 to 4.36) | 3.07*** (2.23 to 4.23) |
| Abuse* second quintile | | | 1.38 (0.75 to 2.55) | | | 1.79 (0.68 to 4.72) | | | 1.32 (0.59 to 2.95) |
| Abuse* middle quintile | | | 2.90** (1.30 to 6.44) | | | 3.83* (1.07 to 13.69) | | | 2.72 (0.93 to 7.91) |
| Abuse* fourth quintile | | | 4.31*** (1.87 to 9.92) | | | 3.36 (0.93 to 12.18) | | | 5.93** (1.87 to 18.81) |
| Abuse* highest quintile | | | 4.47** (1.67 to 11.98) | | | 3.45 (0.76 to 15.74) | | | 5.61* (1.40 to 22.50) |
| Abuse* high education | | | 0.74 (0.32 to 1.70) | | | 0.65 (0.22 to 1.89) | | | 0.78 (0.17 to 3.51) |

Continued

**Table 2** Continued

| Variables | Total | | | Men | | | Women | | |
|---|---|---|---|---|---|---|---|---|---|
| | Model 1 | Model 2 | Model 3 | Model 1 | Model 2 | Model 3 | Model 1 | Model 2 | Model 3 |
| Abuse* have someone to trust | | | 0.87 (0.44 to 1.71) | | | 0.72 (0.22 to 1.89) | | | 0.90 (0.38 to 2.15) |
| Abuse* feel able to manage situations some time | | | 0.92 (0.54 to 1.58) | | | 1.26 (0.53 to 2.98) | | | 0.80 (0.39 to 1.60) |
| Abuse* feel able to manage situations most time | | | 1.04 (0.42 to 2.54) | | | 2.43 (0.75 to 7.90) | | | 0.41 (0.10 to 1.78) |
| Constant | 0.66*** | 0.60* | 0.61* | 0.55*** | 0.51 | 0.52 | 0.77*** | 0.84 | 0.88 |
| Cox & Snell R² | 0.008 | 0.327 | 0.328 | 0.005 | 0.341 | 0.342 | 0.011 | 0.317 | 0.319 |
| Nagelkerke R² | 0.011 | 0.441 | 0.442 | 0.007 | 0.467 | 0.469 | 0.014 | 0.424 | 0.427 |

Source: Authors' analysis of UNFPA Building Knowledge Base on Ageing in India 2011 survey.
Values in the brackets are 95% CIs.
*p<0.05; **p<0.01; ***p<0.001.
GHQ-12, 12-item version of the General Health Questionnaire; UNFPA, United Nations Population Fund.

represented in this research, elder abuse shows a negative association with the mental health of older adults, especially among women; (2) household wealth generally has an inverse relationship with mental health; (3) however, the negative association between elder abuse and mental health is stronger among older people living in wealthy households.

The number of older people in India is steadily growing.[4] Increased life expectancy brings with it more chronic health problems and functional limitations that require long-term care. Most older people continue to live in villages and to experience poor socioeconomic status and are dependent on their families for both financial and physical support. While the need for care has grown, available resources have decreased. The lack of jobs close to where they live forces many young Indians to seek employment in urban areas. Such migration reduces the number of available caregivers and increases the demands on non-migrant family members to shoulder responsibility for elders' care.[2] The costs of care were often high due to a lack of adequate public healthcare for older persons. Mistreatment of one's older parents may also emanate from conflict over the control of family property. Researchers have begun to argue that traditional Indian cultural values and the consequences of urbanisation and modernisation influence the nature and scope of elder abuse.[38] In addition, recent research has highlighted the increasing incidence of elder abuse due to property separation/division, living conditions and the growing generation differences in thinking and attitude towards expectations and lifestyles.[53] As India continues on its path of economic development, with increasing urbanisation and spatial mobility, older people may be further exposed to abuse.

At present, mental health in later life is not a priority area in many low-income countries[54] and how it is associated with elder abuse is neglected in both the research and policy arenas. Intergenerational relations between older people and their adult children are pivotal in the health and well-being of older people. However, such relations can have both positive and negative impacts on the mental well-being of both the older person and, in certain cases, that of the adult child carer.[55-56] Elder abuse needs to be recognised as a key public health issue and appropriate strategies, policies and practices put in place. Reducing elder abuse will have a positive impact on both the physical and mental health outcomes in later life. Policy-makers in India are faced with a major challenge in a low resource context; however, widening the public policy debate to include the recognition of the prevalence of elder abuse and how best to address it within health policy planning would be a key move forward.

**Contributors**

All authors Maria Evandrou, Jane Falkingham, Min Qin and Athina Vlachantoni contributed to conceptualising, writing, and editing this paper, and have read and approved the final manuscript. Data analysis was performed by Min Qin.

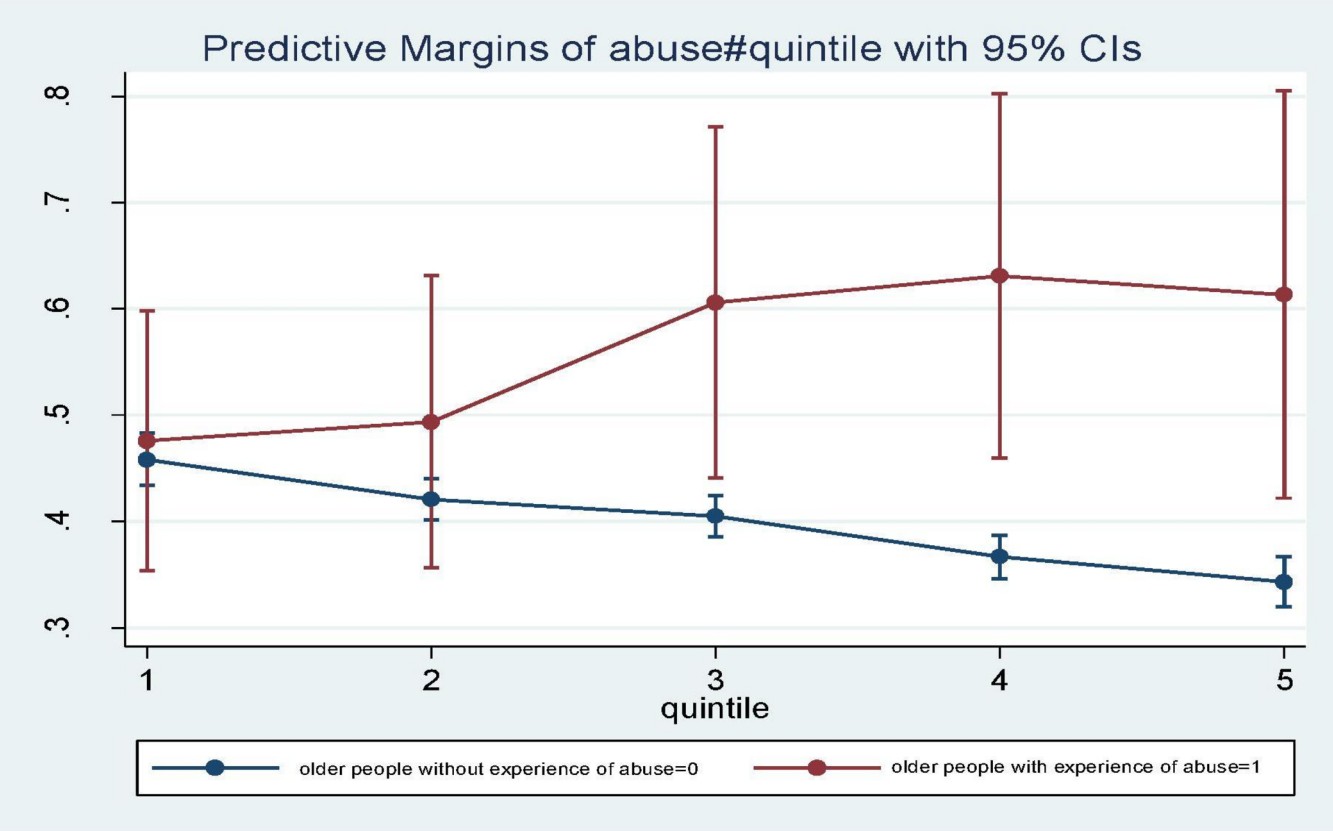

**Figure 1**  Predicted probability of GHQ-12>=4 among older adults by elder abuse and household wealth quintile index. Source: Authors' analysis of UNFPA Building Knowledge Base on Ageing in India 2011 survey. GHQ-12, 12-item version of the General Health Questionnaire; UNFPA, United Nations Population Fund.

**Contributors**  All authors ME, JF, MQ and AV contributed to conceptualising, writing, editing this paper and have read and approved the final manuscript. Data analysis was performed by MQ.

**Funding**  The authors wish to acknowledge the support of colleagues in the ESRC Centre for Population Change No. RES-625-28-0001 and ES/K007394/1, and the ESRC GCRF Global Ageing and Long-Term Care Network (GALNet) No.ES/P006779/1.

**Competing interests**  None declared.

**Ethics approval**  The Ethics Committee in the University of Southampton.

**Provenance and peer review**  Not commissioned; externally peer reviewed.

**Data sharing statement**  No additional data are available.

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
