## [Reviewer comments · BMJ Open]

ARTICLE DETAILS

TITLE (PROVISIONAL)	Elder abuse as a risk factor for psychological distress among older adults in India: a cross-sectional study
AUTHORS	Evandrou, Maria; Falkingham, J; Qin, Min ; Vlachantoni, Athina

VERSION 1 – REVIEW

REVIEWER	Dr. Mala Kapur Shankardass Associate Professor, Maitreyi College, South Campus, University of Delhi, Chanakyapuri, New Delhi 110021, India.
REVIEW RETURNED	09-May-2017

GENERAL COMMENTS	It is an interesting study using data from UNFPA study and authors should explore it further by new data which will appear in due course with national scale longitudinal ageing study in India (LASI) being coordinated by the International Institute for Population Sciences in collaboration with Harvard School of Public Health and University of Southern California, USA.
---

REVIEWER	T.S.Syamala Institute for social and economic change India None
REVIEW RETURNED	24-May-2017

GENERAL COMMENTS	This paper has attempted to explore a dimension of life all will arrive if longevity permits and to address the relationship between elder abuse in India and the consequential psychological distress. Besides, it has also ascertained whether such an association varies with the levels of psychological and material resources. The data for this study is from a representative sample survey conducted by UNFPA and its partners in seven states of India, namely, West Bengal, Tamil Nadu, Punjab, Odisha, Maharashtra, Kerala and Himachal Pradesh. These data-sets provide information on the extent of psychological distress among the older adults and their exposure to abuse. Since the data also have information on psychosocial and material resources, these provide a unique opportunity to examine the relationship between elder abuse and its association with mental health status of the older adults. The paper also brings forth the fact that the older adults from certain states like West Bengal, Odisha and Maharashtra have higher levels of psychological distress.
---

My specific comments are as follows:

1. The GHQ-12 in the BKPAI data used Likert scale with score range of 0-36 and cut off value 12. However, this paper has used a bimodal score with its cut off at 4. With this change in scoring pattern, the levels of psychological distress across states do not match with the published data. Therefore, it is suggested to use the Likert scoring pattern so that there are no variations in the levels.

2. The paper uses elder abuse data for the last one month which I feel is problematic while linking it with the psychological distress as there could also be a time lag between exposures to abuse and its impact on their psychological well-being. Further, there is a possibility that those older persons who would have not faced any abuse during the last month might have faced abuse any time after turning 60. The exposure to abuse anytime during the life course might affect their psychological well-being. Since UNFPA data provides information on abuse after age 60, it is better use that data rather than restricting the analysis merely to the abuse faced during the last month.

3. It may not also be a good idea to limit the abuse perpetuated only by family members. Even abuse by non-family members could also reflect the changing family dynamics in India. An increase in living alone for instance can indicate changes in the family dynamics and the major perpetrators of abuse for those who live alone could be the neighbours.

4. In the list of explanatory variables, instead of using income categorised as, 'no income', 'one source' or 'multiple sources', it is better use the economic dependency status of the older persons because this variable truly indicates the economic dependency of the older persons. Even if they have some sources of income, that does not guarantee economic independence. Many older persons, for example, may be receiving old-age pensions but the amount may be too meagre to make them economically independent.

5. Further, there is need to scrutinise the references closely for its correctness. For example, Reference No 24 has been authored by Moneer Alam, K. S. James, G. Giridhar, K. M. Sathyanarayana, Sanjay Kumar, S. Siva Raju, T. S. Syamala, Lekha Subaiya, Dhananjay W. Bansod and published by UNFPA. Uniformity should also be maintained by the style of referencing in the text by indicating either by author or by number,

After incorporation of these variables in this model, this paper would be a good addition to the existing knowledge on this topic.

This is a well written paper which has put interesting hypotheses to test. It is concise and gives a strong message on this relationship and hence I have no hesitation in recommending it for publication.

REVIEWER	Maria Gabriella Melchiorre - Senior researcher INRCA-IRCCS (National Institute of Health and Science on Ageing) - Italy
REVIEW RETURNED	02-Jul-2017

GENERAL COMMENTS	This study presents the results of primary scientific research. It's very interesting and well written. The research questions are well formulated. Methods are very detailed and exhaustive. Anyway, there are some criticisms to be addressed, that is some adjustments/integrations and explanations from authors are needed, as I've described in the sections of the review below. General aspects I notice that authors use different terms to indicate (apparently) the same issue, i.e.:  - "psychological distress" in the title of the paper, and in the "Measurements" and "Results" sections of the paper; - "mental health" in the "Discussion and conclusion" section of the paper, in the "Results" and "Conclusions" sections of the Abstract, and among the key words. I think authors should explain better this aspect, in order to clarify how they use both terms, e.g. as synonyms (as it seems). Authors moreover state "Mental health" as the secondary subject heading (the first one being "Public health"). Furthermore, authors should check the use of the terms "per cent" and "percent", in order to decide which of them they want to use along the whole paper. Sections of the manuscript: Abstract: I suggest to articulate the sub-section "Methods" as follows:  - "Design", .e.g. cross-sectional study; - "Setting", e.g. data come from a representative sample of 9,692 adults aged 60 and above in seven Indian states...; I also would suggest to indicate the final sample of 9.589 adults, as it is also in Table 1 as number of respondents (instead of the total sample size interviewed of 9.692). - "Methods", e.g. data come from the UNFPA project Building Knowledge Base on Ageing in India (BKPAI), which was conducted in 2011. Any reported elder abuse (physical and/or emotional) from family members one month before the survey was examined. Key Words: I suggest to add "psychological distress" and "India". Introduction: This section is well done. I suggest anyway some additional reference on EA in general, not only in India, as more wide background for the issue, e.g. the following:  - Yon Y. et al., Elder abuse prevalence in community settings: a systematic review and meta-analysis, Lancet Glob Health 2017;5: e147–56. http://thelancet.com/pdfs/journals/langlo/PIIS2214-109X(17)30006-2.pdf - Pillemer K. et al., Elder Abuse: Global Situation, Risk Factors, and Prevention Strategies, Gerontologist, 2016, Vol. 56, No. S2, S194–S205. https://academic.oup.com/gerontologist/article-lookup/doi/10.1093/geront/gnw004
--

- Dong XQ., Elder Abuse: Systematic Review and Implications for Practice, J Am Geriatr Soc. 2015 Jun;63(6):1214-38.

<http://onlinelibrary.wiley.com/doi/10.1111/jgs.13454/pdf>

I also would suggest to move the sentence "This study aims to contribute to the literature by investigating the association between elder abuse and psychological distress among older adults in India and examining whether this association varies by the level of psychosocial and material resources at older adults' disposal" from p. 5 to the end of the section "The relationship between elder abuse, resources and psychological distress".

Methods

I think authors should distinguish better the sub-sections (e.g. Data, Measurements, Statistical analysis) from the "Methods" heading, for instance by writing this in capital letters.

Moreover, I suggest the following:

- probably "Elder abuse" could be the first measure listed in the section "Measurements", followed by "Psychological distress" and "Psychosocial and material resources";

- when authors describe "Material resources" (p. 11), they could add the term "index" close to the one "quintile", in order to harmonize the description of "household wealth" in this section and in Tables 1 and 2;

- I would like to see references in the paper concerning both measures "ADLs" and "Disability". Also authors should clarify better the difference between these two dimensions (as they were used in their study);

- finally, the sub-section "Method" should be "Statistical analysis", and also it should first refer to bivariate analyses (as stated in the abstract), and include information on both the significance level set for the analyses (e.g. $p < 0.05$, or $0.01 \dots$) and the statistical packages (e.g. SPSS or STATA) that was used to carry out the analyses themselves.

Ethics approval

Authors state (p. 20) that ethical approval for the study has been obtained from the Ethics Committee in the University of Southampton. Authors also state (in the sub-section "Elder abuse", p. 9) that careful attention was paid to avoid the appearance of any family members during this particular part of data collection and to guarantee the confidentiality of information. Anyway, I suggest to include a separate sub-section on ethics in the paper (under the section "Methods"), where additional/more detailed information is reported, e.g.: how were participants informed (verbally? By written format?) about the aims and methods of the survey before the interview; how were participants assured regarding their voluntary participation and anonymity/privacy/confidentiality of collected information; were participants informed on the use of the collected data/information, e.g. only for the purpose of the study?; was informed (written?) consent obtained from each participant in the study?

Results

Authors should on the whole specify better the direction of the associations, e.g.:

- p.14: Indicators of health status including fair/poor self-related health, heavy difficulty with ADLs and disability all show a positive association with psychological distress;

- p.15: Social activity participation, social support (having someone to trust) and mastery (feeling able to manage situations) are all associated with psychological distress in the expected negative direction;

- p.15: The results indicate that psychosocial resources only have a direct negative association with psychological distress. Authors could also include some numbers from Table 2 in order to explain/support better their considerations. Moreover, authors should include in the "Results" section some findings that are then discussed in the "Discussion" section, e.g.:

- p. 17: Interestingly the results in this paper show that household wealth has a direct and inverse relationship with psychological distress...

- p.17: results also add to the evidence base with regard to inequalities in older people's mental health related to levels of physical health and different geographic areas....
Finally, authors indicate Figure 2 instead of Figure 1 in the paper (end of p. 15).

Discussion and conclusion

First of all I would suggest to keep separated "Discussion" from "Conclusion". By my opinion, this paragraph could be as follows:

- Discussion: from p. 16 to p. 18 (first two lines);

- Limitations: lines 23-46 p. 18;

- Conclusion: lines 10-20 p. 18, lines 49-57 p. 18, and pp. 19-20. Moreover:

- The first sentence of Discussion could be specified as follows: The analyses in this paper suggest that elder abuse has a significant negative association with the mental health of older Indians;

- Authors should also specify better when findings are from their study or from previous literature, e.g.:

o p. 16: However, no evidence (from their study I suppose) is found for buffering effects of psychosocial resources, such as social support and perceived ability to control outcomes;

o p. 16: The results (from their study I suppose) only highlight a direct and beneficial association...

o p.17: The results (from their study I suppose) also add to the evidence base with regard to inequalities...

Tables

In both Tables it could be better to state or "sex" or "gender". Currently I see "sex" in Table 1 and "gender" in Table 2; moreover, in both Tables it could be better to state "Difficulty with ADLs".

Currently in Table 1 I see only "ADLs". Finally, I would suggest to integrate Odds ratios (OR) reported in Table 2 with values concerning Confidence Intervals (CI), and the indications (as footnote) of what the number of asterisks indicate (e.g. I suppose they are the following p-values: * $p < 0.001$, ** $p < 0.01$, *** $p < 0.001$).

REVIEWER	Jutta Lindert, Professor of Public Health University of Applied Sciences Emden, Germany None
REVIEW RETURNED	10-Jul-2017

GENERAL COMMENTS	The focus of this paper is very interesting and necessary. We need more data from more countries to better understand elder abuse. However, the measures used might need reconsideration. Additionally, the theoretical background has room for improvement. Specifically interesting might be perceptions of abuse and how abuse is defined in India, taking into account gender and various groups. I was wondering whether it is possible to Analyse the different groups in India together as one sample. Might it be possible to Analyse these groups in separate analyses? Likewise, theories of elder abuse which have been elaborated in the past years might be considered en detail. Taken together focusing on India and mental health problems in India is promising for further research.
--

VERSION 1 – AUTHOR RESPONSE

Reviewer 1

Response: We are pleased that Reviewer 1 thought that this was an interesting study.

Reviewer 1 suggests that the team might explore this issue further by applying for access to the forthcoming new longitudinal ageing study in India (LASI) dataset. We thank the Reviewer for this useful suggestion, and plan to apply to use the data once it becomes publically available.

Reviewer 2

Responses: We thank Reviewer 2 for their very helpful set of suggestions and address each of these in turn below, using the numbering in the Reviewer's own comments for ease of reference.

1. Reviewer 2 usefully highlights that the report published on the BKPAI data used the Likert scale with a score range of 0-36, and a cut off value of 12 to measure psychological distress and suggested that we use the same approach in order to be in line with the published survey report. We thank the reviewer for this suggestion. We did originally consider using this approach, as both scoring methods i.e. the GHQ-12 method used in the paper (0-0-1-1) and the Likert scale method (0-1-2-3) used by the BKPAI team are widely used. However, we were guided in our choice by the results of a previous study comparing the two approaches (Goldberg et al, 1997), which concluded that the GHQ-12 method performed better. This decision was reinforced by the fact that the majority of studies in the recent past have also used the GHQ-12 method.

Following receipt of Reviewer 2's comments, we have conducted some comparisons using both scoring methods. Amongst those identified as distressed cases using the GHQ-12 (0-0-1-1) scoring method with 4 as the cut-off score, 95% were also identified as distressed cases using the Likert scale scoring method. However, amongst those identified as distressed cases using the Likert scale scoring method, only 83% were identified as the distressed cases using GHQ-12 scoring method (data not shown).

The results (Table A1 submitted as the supplementary file to the editor only) show that the prevalence of psychological distress is 46.6% using the Likert scale method, which is somewhat higher than the 40.6% using the GHQ-12 method. However, the patterns of associations of psychological distress with exposures and potential risk factors are similar for both scoring methods. Thus, after careful consideration, we have decided to continue to use the GHQ-12 scoring method, with the cut-off score of 4 in the main analyses in the paper. We have added a sentence (p.11) to refer to the Goldman et al, 1997 study.

2 & 3. Reviewer 2 usefully highlights that the BKPAI dataset includes information on the experience of any abuse after age 60 and suggests that we consider using this information (comment 2). They also suggest we consider including abusive cases by non-family members such as neighbours (comment 3) given the trend towards living alone. We thank Reviewer 2 for these suggestions.

As outlined in our introduction (p4-5), a key motivation for the paper is to examine elder abuse in the context of wider demographic and socio-economic transformations that are affecting the traditional Indian family system. Our decision to limit abuse to that reported as emanating from family members was a deliberate choice, as it is this form of abuse that we hypothesise may have increased as a result of recent changes such as increasing female labour force participation, the migration of adult children etc. Moreover, although we recognise that an increasing proportion of older people in India live alone, from the survey report and from previous studies, it remains the case that the major perpetrators of older people abuse in India are family members (Dyer et al 2000), including family members living outside the household.

Information regarding the source of abuse i.e. family member or others is only available for those older people who report experiencing abuse in the last month and hence our decision to use this information, rather than widening our focus to all instances of abuse post age 60. Focussing on abuse in the last month also has the added advantage that it allows us to the contemporaneous interaction between elder abuse, psychosocial and material resources and psychological distress.

4. Reviewer 2 usefully recommends the use 'economic dependency status' as a control variable as an alternative to the 'number of income sources'. Thank you for this suggestion; we have taken this helpful advice and replaced the variable on income sources with the new variable capturing the financial dependency of the older person (pages 12, 31, 33; and Tables 1 and 2).

5. Reviewer 2 helpfully highlighted the need to further check the references. We have done this and made the appropriate revisions. In particular, we have changed the authorship of the BKPAI report from UNFPA to the contributors Alam M, James KS, Giridhar G, et al (2012) (p.27).

Reviewer 3

We are pleased that Reviewer 3 thought that this was an interesting study and thank them for their detailed comments, which we have taken on board and feel that they have helped to improve the paper. The revisions in response to their comments are detailed below.

Response:

- In the revised Abstract (p.2), we have revised the material under the previous "Methods" heading into separate sections on "Design"; "Setting" and "Methods".
- We have also included "psychological distress" and "India" in the keywords. Thank you for the suggestions.
- Throughout the paper, 'percent' has been used.
- In the revised INTRODUCTION, additional information on elder abuse worldwide has been included as a broader context, with all three of the papers suggested by Reviewer 3 now referenced (pages 5, 24-25; references 8-10).

- The revised paper now explains the relationship between psychological distress, mental health and public health in the beginning of the section entitled 'The relationship between elder abuse, resources and psychological distress' (p.6). The aims of the study are also clearly spelt out at the end of this section (p.8).
- In the revised DATA & METHODS section, the sub-sections 'Measurements' and "Statistical analyses" are clearly now distinguished.
- As suggested by Reviewer 3, in the "Measurements" sub-section we have revised the order in which the variables are discussed, with "Elder abuse" now being the first measure (p.9) followed by "Psychological distress" and "Psychosocial and material resources". The term "index" has been added to the discussion of the "household wealth quintile" (p.12).
- Reference 38 on the measures of "ADLs" and "Disability" has been included in the paper and in the references. The difference between these two dimensions of the limitations of functionality has been clarified on p. 13. ADLs refer to the ability to perform basic daily activities; disability is associated with a decline of motor function. Usually, a high disability score represents a lower ability to perform ADLs (p.13).
- The sub-section "Statistical analyses" (p. 14), includes information of the statistical packages STATA which was used to conduct the analyses.
- A separate sub-section on Ethics approval (p.15) has been included, with further information about informed consent and anonymity/privacy/confidentiality of the data collected.
- The RESULTS section has been revised to include data from Table 2 (relevant p-values and test statistics) to explain/support our statements, and the direction of the associations has been more clearly specified (pages 15-18).
- In addition, we have included discussion of the association between household wealth, physical health and different geographic areas with psychological distress.
- We have corrected the typo on Figure 1 (p.18).
- The previous 'Discussion and conclusion' section has now been revised and separated into three separate sections entitled DISCUSSION, LIMITATIONS and CONCLUSION (pages 18-22).
- The first sentence of the Discussion clarifies that 'The analyses in this paper suggest that elder abuse has a significant negative association with the mental health of older Indians'. We have also clearly specified where findings are from this study and where they are from other literature.
- We have presented in Table 2 (p.33) both Odds ratios (OR) and 95% Confidence Intervals (CI), with the indications of p-values: * $p < 0.001$, ** $p < 0.01$, *** $p < 0.001$. We have revised both Tables to state "Gender" and "Difficulty with ADLs" (p.31).

Reviewer4

We are pleased that Reviewer 4 thought that this was an interesting and necessary study and thank them for their comments.

Responses:

In response to Reviewer4's suggestion to strengthen the theoretical background of elder abuse and perceptions about abuse in the Indian context, we have included a brief discussion of three theoretical perspectives (the social exchange theory, situational theory and symbolic interactionism theory) which help to explain elder abuse by the family members (p.5). We also discussed perceptions about abuse in the Indian context based on qualitative studies (p.5).

Reviewer 4 very helpfully suggested analysing different groups, such as by gender. We have taken this suggestion on board and have run separate logistic models for the whole sample (with gender as a control), and then among older men and older women. The corresponding results are discussed (pages 15, 17, 18, 19, 21).

Reviewer 4 suggested referring to more data from a wider range of countries and a reconsideration of the elder abuse measures. In response, we have included some additional information on elder abuse worldwide in the INTRODUCTION, as part of setting the research in a broader context (p.5). We have also included a discussion of the measurement of elder abuse, given the restrictions of secondary data analysis, in the Limitations section (p.20).

VERSION 2 – REVIEW

REVIEWER	Maria Gabriella Melchiorre - Senior researcher INRCA – National Institute of Health and Sciences on Ageing, Italy
REVIEW RETURNED	03-Sep-2017
GENERAL COMMENTS	Elder abuse as a risk factor for psychological distress among older adults in India: a cross-sectional study The Authors did a great work in revising the manuscript. They carefully solved all the criticisms I highlighted in my previous review. I've only 3 little final suggestions, as follows:  1) the paragraph "The relationship between elder abuse, resources and psychological distress" seems a sub-section of the INTRODUCTION. Thus I think it should not be written in capital letters; 2) the sub-section "Statistical analysis" could include information on the significance level set for the analyses (e.g. $p < 0.05$); 3) in the paragraph "Ethics approval" Authors could specify if the informed consent was written or verbal.